# Analysis of Receptor-Type Protein Tyrosine Phosphatase Extracellular Regions with Insights from AlphaFold

**DOI:** 10.3390/ijms25020820

**Published:** 2024-01-09

**Authors:** Lina El Badaoui, Alastair J. Barr

**Affiliations:** School of Life Sciences, University of Westminster, 115 New Cavendish Street, London W1W 6UW, UK; l.elbadaoui@westminster.ac.uk

**Keywords:** receptor-type protein tyrosine phosphatases (RPTPs), CD148, R3 subgroup, extracellular domain (ECD), fibronectin, polymorphisms, AlphaFold

## Abstract

The receptor-type protein tyrosine phosphatases (RPTPs) are involved in a wide variety of physiological functions which are mediated via their diverse extracellular regions. They play key roles in cell–cell contacts, bind various ligands and are regulated by dimerization and other processes. Depending on the subgroup, they have been described as everything from ‘rigid rods’ to ‘floppy tentacles’. Here, we review current experimental structural knowledge on the extracellular region of RPTPs and draw on AlphaFold structural predictions to provide further insights into structure and function of these cellular signalling molecules, which are often mutated in disease and are recognised as drug targets. In agreement with experimental data, AlphaFold predicted structures for extracellular regions of R1, and R2B subgroup RPTPs have an extended conformation, whereas R2B RPTPs are twisted, reflecting their high flexibility. For the R3 PTPs, AlphaFold predicts that members of this subgroup adopt an extended conformation while others are twisted, and that certain members, such as CD148, have one or more large, disordered loop regions in place of fibronectin type 3 domains suggested by sequence analysis.

## 1. Introduction

Protein tyrosine phosphorylation is a post-translational modification that plays a crucial role in signal transduction of multiple cellular processes relevant to immunology, haematology, metabolism, neurobiology, as well as in disease [1]. Protein tyrosine phosphatases (PTPs) work in concert with protein tyrosine kinases (PTKs) to regulate tyrosine phosphorylation and it has been suggested that protein phosphatases have a more pronounced effect on rate and duration of signalling events while kinases control amplitude [2]. The 38 ‘classical’, tyrosine-specific, human PTPs are classified into 17 intracellular (non-receptor) PTPs and 21 receptor-type PTPs (RPTPs) with diverse extracellular modules which can be further sub-divided into 8 subgroups (R1–R8) based on their domain compositions [3,4,5] (Figure 1). The RPTPs have either a single highly conserved intracellular phosphatase domain or a tandem arrangement of phosphatase domains in which the membrane-proximal (D1) domain has catalytic activity and the distal D2 domain is either inactive or has low-level activity. Given their important role in signalling many mechanisms have been uncovered that regulate phosphatase activity and these mechanisms often involve RPTP extracellular domains. These mechanisms include ligands, dimerization, oligomerization, restricted sub-cellular localisation, post-translational modification, oxidation, and proteolytic processing [6,7,8,9]. Understanding the detailed structural features of RPTPs is important since these signalling molecules are often mutated in disease. In addition, targeting the extracellular region of RPTPs as a therapeutic strategy offers an alternative to the challenge of inhibiting the phosphatase activity [10]. This review provides an overview of recent insights into RPTP structure and function and examines predicted structures of RPTPs generated by AlphaFold, a state-of-the-art machine learning model from DeepMind, which has been shown to generate highly accurate three-dimensional (3D) structures of proteins from their amino acid sequences [11,12].

## 2. Subgroup R1/R6 (CD45)

CD45 is expressed on almost all haematopoietic cells, except erythrocytes, and plays a crucial role in both T-cell and B-cell antigen receptor signalling. Alternative splicing of CD45 gene transcripts gives rise to several isoforms that are expressed in a cell-type-specific manner, and which have a wide variation in the highly glycosylated extracellular domain. Upon interaction of the T-cell with the antigen presenting cell (APC) at the immunological synapse, CD45 is excluded from this region of close cell–cell contacts where T cell receptors (TCRs) accumulate, and it is this separation of TCRs from CD45 that is thought to shift the equilibrium to phosphorylation of immune receptor tyrosine-based activation motifs (ITAMs) triggering TCR activation. One model explaining the molecular mechanism for this separation is known as the kinetic-segregation (K-S) model [13], and the extracellular domain of CD45 is key to the process. The extracellular domain of CD45 is composed of a mucin-like region, which is spliced out in some isoforms, followed by a cysteine-rich domain and three fibronectin type 3 (FN3) domains. Structural studies have determined that this region extending directly out from the cell membrane is rigid, and its size (~25–50 nm) would account for it being excluded from the narrow (13 nm) gap between the TCR and APC (reviewed in [14]). The rigid arrangement of FN3 domains is determined by very short inter-domain interfaces that allow extensive interdomain interactions and by cysteines forming disulfide bridges [15]. The rigid arrangement is comparable to that observed in RPTPμ of the R2B subgroup but contrasts with the highly flexible RPTPσ of the R2A subgroup where domain interfaces are smaller or nonexistent (see below). AlphaFold structure predictions correlate closely with the conclusions from experimentally determined structures and indicate that these characteristics are shared by subgroup members (Figure 2).

In addition to the involvement of the CD45 extracellular domain in size-based exclusion, it also binds ligands called galectins. Galectins are a family of soluble proteins with multiple biological activities that specifically bind galactose containing glycans [16].

## 3. Subgroup R2A (RPTPσ, RPTPδ and Leukocyte Common Antigen-Related (LAR))

Receptor PTPs of the R2A subgroup are characterized by an extracellular domain composed of three N-terminal immunoglobulin (Ig)-like domains, followed by five to nine FN3 domains, which may be modified by alternative splicing, a transmembrane domain and tandem arrangement of intracellular phosphatase domains. A wide range of ligands have been described for these RPTPs with roles in neuronal growth and the formation of synapses [9]. In addition to a role within the nervous system, involvement in arthritis and autoimmune events has been reported [10]. The ligands include both protein and proteoglycan molecules. Early work in *Drosophila* showed that the heparin sulfate proteoglycans (HSPGs) syndecan and dallylike are *Drosophila* LAR ligands involved in controlling formation of the neuromuscular junction [17]. Other ligands include chondroitin sulfate proteoglycans (CSPGs), Tyrosine Kinase Receptor C (TrkC), and several others [7,9] and Netrin-G ligand-3 (NGL-3) which interacts with all three R2A RPTPs and is involved in the formation of excitatory synapses [18,19].

Structural studies and biophysical studies have demonstrated that, in contrast with CD45, the extracellular domain of RPTPσ is flexible allowing interaction with ligands either within the same cell (*cis*) or in an adjacent cell (*trans*) [20,21]. Interaction with HSPGs and CSPGs involves exclusively the glycosaminoglycan chain interacting with the first Ig domain. This region overlaps with the TrkC binding region, which involves the first three Ig domains, while in contrast, NGL-3 binds to the FN3 repeats of the R2A PTPs. Analysis of experimental structures for the ectodomain of PTPRσ and PTPRδ (PDB: 4PBX, 4YH7) shows that Ig1 and Ig2 domains interact tightly to form a V-shape; there is then a hinge between Ig2 and Ig3 followed by FN3-1 and FN3-2 in a straight line and FN3-3 in a perpendicular orientation. Superimposition of the first four FN3 domains of LAR (PDB: 6TPW) indicates that its L-shaped structure places the FN3-3 and FN3-4 domains in an orientation almost at 180° relative to FN3-3 of PTPRσ and PTPRδ, indicating the flexibility of the linker between FN3-2 and FN3-3 for the R2A subgroup PTPs. These observations are supported by the twisted conformations of the AlphaFold structure predictions for the subgroup (Figure 2).

It has been noted that although HSPGs and CSPGs bind to a common site on PTPRσ they mediate distinct physiological outcomes, activation versus inhibition of neuronal growth, respectively. This has been explained by the ability of HSPGs expressed on the same cell as PTPRσ to cause clustering of PTPRσ away from potential substrates thereby enhancing the duration of signalling due to phosphorylated proteins. On the other hand, interaction of PTPRσ with CSPGs expressed on a different cell would maintain the RPTP in its monomeric and active state. The flexible and twisted AlphaFold structure predictions support this model (Figure 2). Related to this model, an exciting recent study has developed a monoclonal antibody targeting the extracellular domain of PTPRδ which induces dimerization, consequently inhibiting of phosphatase activity, which has therapeutic implications for breast cancer [22].

## 4. Subgroup R2B (RPTPμ, RPTPρ, RPTPκ, and PCP2/RPTPλ)

Subgroup R2B RPTPs are notable for undergoing homophilic interactions that mediate cell–cell contact [7,23]. The extracellular domain of these RPTPs contains one meprin-A5 antigen-PTP mu (MAM) domain, an Ig-like domain (Ig) and four FN3 repeats followed by a transmembrane domain and cadherin-like sequence and tandem arrangement of intracellular phosphatase domains as with the R2A subgroup. Recent work has determined that PTPRλ is unique within the family in that both domains are catalytically inactive pseudo-phosphatase domains [24]. Structural studies have revealed that the MAM and Ig domains are tightly associated to form one unit, and it is this region together with the first and second FN3 domains that forms the interaction surface involved in homophilic interactions [25]. Analysis of the RPTPμ structure indicates that the ectodomain forms a rigid rod with the *trans* homophilic interactions setting a specific inter-cellular distance. Interestingly, in contrast to the situation described above for CD45 and TCR signalling, the distance matches closely with that formed by cadherin ectodomain *trans* dimers, with which RPTPμ is known to associate. Other members of the R2B family, except for PTPRλ, mediate cell–cell interactions, although, despite the high cross-family sequence similarity, they do not form heterophilic interactions. A recent comparison of two members of the family (RPTPμ and RPTPκ) revealed that homophilic specificity is determined by a combination of shape complementarity and key sequence differences [26]. Predicted AlphaFold structures are consistent with the experimental structural data and show that all R2B subgroup members adopt the rigid rod structural conformation in contrast with the highly flexible R2B subgroup (Figure 2).

In addition to a role in developmental processes, a large body of literature provides evidence for involvement of this subgroup of RPTPs in cancer. Depending on the subtype, they have been classified as tumour suppressors and oncogenes in various cancers [23,27].

## 5. Subgroup R3 (CD148, VE-PTP, GLEPP1, SAP-1 and PTPRQ) 

There are five mammalian receptor PTPs in the R3 subgroup, CD148 (also known as DEP-1, *PTPRJ*), vascular endothelial protein tyrosine phosphatase (VE-PTP, *PTPRB*), glomerular epithelial protein 1 (GLEPP1, *PTPRO*), stomach cancer–associated protein tyrosine phosphatase-1 (SAP-1, *PTPRH*) and PTPRQ, which are characterized by an extracellular domain formed solely of multiple FN3 domains followed by a transmembrane domain and a single intracellular phosphatase domain [28]. The phosphatases alldephosphorylate phospho-tyrosine residues, with the exception of PTPRQ which is a phosphatidylinositol phosphatase and has only weak activity towards protein substrates [29]. The predicted number of FN3 domains in the human R3 PTPs ranges from 9 to 17, based on UniProt annotations, and they are modified by N-linked glycosylation. Various splice variants have been described for this subgroup of RPTPs. Alternative splicing of the PTPRJ gene produces a soluble form of CD148 (sPTPRJ) lacking the transmembrane domain and intracellular phosphatase domain [30], although the functional significance of this protein is unclear. Splicing of the PTPRO gene produces a truncated isoform (PTPROt) that lacks the extracellular domain but has a transmembrane domain and intracellular phosphatase domain that is identical with full-length PTPRO. These PTPRO isoforms show tissue-specific expression with PTPROt found preferentially in haematopoietic cells [31,32] while the full-length PTPRO is expressed in kidney and brain.

A role for the extracellular domain of specific R3 subgroup PTPs in ligand binding, dimerization and size-exclusion from regions of close cell–cell contact, as discussed for CD45 (see above), has been reported; however, structural information is lacking for the extracellular domain of this subgroup and this region is poorly understood.

### 5.1. Ligands for R3 PTPs and Interactions Mediated via the Extracellular Domain

Several ligands have been reported for CD148; yet, to our knowledge, R3 subgroup ligands for other members of the have not been documented despite several decades of research in this area. This raises the questions: Are these other R3 subgroup RPTPs orphan receptors whose endogenous ligands have not yet been identified? What is the function of the large extracellular domain if not involved in ligand interactions, and what features of CD148 determine its ability to interact with ligands? Here, we review current knowledge relevant to these questions and discuss variants of R3 PTPs where a functional effect has been documented. 

Thrombospondin-1 (TSP1) and syndecan-2 (SDC2) were identified as CD148 ligands around the same time [33,34]. Thrombospondins are a family of five secreted modular glycoproteins that bind to the components of the extracellular matrix as well as various cell surface receptors. TSP1 was first identified as a protein released from thrombin-stimulated platelets and interacts with CD36 and CD47 in addition to other receptors [35]. It has a homotrimeric structure of 450 kDa and Takahashi et al. reported that it binds to the extracellular domain of CD148 with high affinity and specificity and increases CD148 phosphatase activity. Follow-on studies have shown that naturally occurring variations in the CD148 extracellular domain Q276P/R326Q mutations do not have major effects on TSP1–CD148 interaction, indicating that these residues are not involved [36,37]. In contrast with the soluble secreted TSP1, syndecans are cell-surface heparan sulfate proteoglycans that regulate many cellular functions. SDC2 has been identified as a ligand for CD148 and is thought to interact in a *trans* manner with CD148 to activate phosphatase activity and promote cell adhesion via integrin β1. Interestingly, a synthetic 18 amino-acid peptide derived from the SDC2 ectodomain (SDC2-pep) is sufficient to bind and activate CD148 and use of this peptide has been proposed as a potential therapeutic strategy for idiopathic pulmonary fibrosis [38].

Vascular endothelial protein tyrosine phosphatase (VE-PTP) shows highly selective expression in endothelial cells, as its name suggests, where it is an important regulator of endothelial tight junctions and vascular permeability (reviewed in [39]). This function is achieved by acting on the membrane protein substrates VE-cadherin, an adhesion molecule, and Tie-2, a tyrosine kinase receptor for angiopoietins. Vascular endothelial growth factor (VEGF) induced tyrosine phosphorylation of VE-cadherin is important for increasing vascular permeability and transmigration of leukocytes while VE-PTP counteracts this process. It has been reported that VE-cadherin interacts directly with the membrane-proximal 17th FN3 domain of VE-PTP [40] and that dissociation of VE-PTP from VE-cadherin increases permeability of the endothelial layer and transmigration. It is interesting to note that this interaction between VE-PTP and its substrate VE-cadherin is, to our knowledge, the only documented example to date of an R3 PTP substrate interaction which involves the extracellular domain. It raises a question as to whether comparable RPTP substrate interactions via the extracellular domain have yet to be identified for the other R3 subgroup PTPs, and the role of the multiple other FN3 repeats.

### 5.2. R3-PTPs: A Role for Size Exclusion

In addition to binding ligands, it has been reported that the extracellular domain of CD148 functions in a similar manner to that described above for CD45, in which the large and structurally rigid extracellular domain is excluded from regions of close cell–cell contact at the immunological synapse where TCRs accumulate. It is this separation of both CD148 and CD45 from TCRs that is thought to shift the balance to phosphorylation of TCRs and activation [41,42]. A close parallel with the role of CD148/CD45 at the immunological synapse has been described at the ‘phagocytic synapse’ where phagocytic immune cells (macrophages, dendritic cells and neutrophils) must be able to distinguish between direct binding to microbes and soluble ligands shed from the surface of microbes [43]. Phagocytes express the transmembrane protein Dectin-1 (also known as CLEC7A), which detects β-glucans in fungal cells triggering anti-microbial activity such as phagocytosis and the rapid release of reactive oxygen species known as the oxidative burst. Dectin-1 binds both soluble β-glucans and particulate forms (such as those presented on the surface of yeast); however, only the particulate form activates dectin-1. This tight regulation of the activation process is required to avoid inadvertent activation of processes that could damage the host, and it is achieved by clustering of the dectin receptor within a region of close cell–cell contact (e.g., neutrophil—yeast), i.e., the phagocytic synapse, from which CD45 and CD148 are excluded by virtue of their large extracellular domain structures. Dectin-1 signals via an ITAM motif, resembling that found in TCRs, and downstream signalling is dependent on tyrosine phosphorylation of the motif so excluding the phosphatases, CD45 and CD148, is necessary for activation to be initiated. In a situation where there is no ligand bound to dectin-1, or a soluble ligand is bound, CD45 and CD148 remain proximal to dectin-1, dephosphorylating the ITAM and consequently preventing activation. Goodridge and colleagues proposed that CD148 may be involved in a similar mechanism of regulation for the C-type lectin family member CLEC2 which has a role in platelet aggregation and is also found on the surface of neutrophils where it is capable of inducing phagocytosis. CLEC2 is a receptor for the membrane protein podoplanin, oxidized hem (hemin) and diesel exhaust particles [44].

### 5.3. Structural Analysis of R3-PTPs Extracellular Domains

For CD148 to function via the size exclusion mechanism described above it requires that the extracellular domain is rigid and extends perpendicular to the plane of the membrane. Currently, structural information for extracellular regions for any of the R3 PTPs is limited to a few CD148 FN3 domain structures and AlphaFold structural predictions (Figure 3).

AlphaFold structures for the extracellular region of R3 subgroup PTPs provide a fascinating insight into the predicted conformation for this subgroup. Both CD148 and SAP1 display an extended conformation with all the FN3 domains in a near linear arrangement; GLEPP1 exhibits a bent inverted L-shape, and both VE-PTP and PTPRQ have a series of linear domains closer to the N-terminus with the others in a twisted conformation. The predicted number of FN3 domains found in some of the AlphaFold structures for this subgroup differs from the number indicated in the corresponding UniProt entry based on PROSITE patterns (Table 1). Additionally, three of the R3-PTPs (CD148, GLEPP1 and PTPRQ) are predicted to have one or more large, disordered loop regions.

The difference between UniProt PROSITE domain predictions and AlphaFold predictions raises the following questions: (1) How confident can we be in the AlphaFold predictions, in particular the reliability of relative positions and orientations of different domains? (2) Are the large, disordered loop regions failures of AlphaFold, i.e., would these regions adopt a well-defined 3D structure in an experimentally determined structure? In regard to these questions, AlphaFold produces a per-residue confidence metric called the predicted local distance difference test (pLDDT) on a scale from 0 to 100, which gives an estimate of how well the prediction would agree with an experimental structure. As expected, the metric gives a higher confidence on domains and lower confidence on linkers and unstructured regions (Figure 4A). 

Another output of AlphaFold is the predicted aligned error (PAE) which measures the confidence in relative position of two residues within the predicted structure, providing insight into the reliability of relative position and orientations of different domains (see https://alphafold.ebi.ac.uk/faq, accessed on 10 November 2023, for more information). The PAE is generally high for residue pairs from different domains of the R3-PTPs indicating that their relative position and orientation is uncertain and should, therefore, be interpreted with caution. It should also be noted that the disordered loops, which are annotated as low confidence predictions, are not necessarily the representative conformation and are likely to exist in dynamic heterogeneous conformations in physiological conditions [45].

Only a few experimentally determined structures of the R3-PTP domains are available and Figure 4 shows a superimposition of those available for CD148 on the CD148 AlphaFold structure. The X-ray crystal structure of CD148 FN3 domains 1 and 2 (PDB: 7U08) superimposed closely onto the AlphaFold structure (RMSD = 1.34 Å over 1050 atoms) as did an NMR solution structure of FN3-4 (PDB: 2DLE) (RMSD = 0.77 Å over 543 atoms). It is interesting to note that the loop sequence is present in the 7U08 construct used for crystallization; however, electron density was missing for this region, suggesting it is indeed flexible. A superimposition of the CD148 AlphaFold structure with a free modelling prediction for CD148 using ColabFold [46], executed without templates, again predicts a loop projecting from FN3-2 of CD148 (Figure 4C).

The functional significance, if any, of the disordered loops predicted in CD148, GLEPP-1 and PTPRQ extracellular domains is unknown; however, it is well established that many polypeptide sequences do not fold into a defined 3D structure yet still carry out their function in a disordered state [47]. These intrinsically disordered regions (IDRs) can achieve this by adopting a specific conformation on binding of an interaction partner, and this offers the advantage of adopting different conformations and enabling interactions with several different interaction partners. These IDR interactions with targets can achieve high specificity and low affinity to enable rapid dissociation of the complex when signalling is complete. One example of such an interaction is the extensively studied binding of the extracellular matrix protein fibronectin with integrins via the RGD sequence in FN3-10 [48].

### 5.4. Genetic Variants of R3 Subgroup PTPs

Numerous genetic variants of PTP genes have been identified that are linked to disease states [49]. Here, we discuss missense single nucleotide polymorphisms (SNPs) in the extracellular domain of CD148 and PTPRQ with documented links to disease, and we map them onto the AlphaFold structures. Studies of patients heterozygous for the Q276P and R326Q alleles of CD148 were found to have platelets that were hypo-responsive to activating stimuli, and this provided a protective effect from heparin-induced thrombocytopenia [50]. The study suggests that the presence of proline and glutamine at positions 276 and 326, respectively, is associated with a loss of CD148-mediated phosphatase activity. Other studies have examined the link between these SNPs and cancer. One meta-analysis of data from 5 studies examining cancer risk linked with these SNPs concluded that they are not associated with an increased risk of cancer except for the R326Q polymorphism in colorectal cancer [51]. The SNPs do not alter cell-surface expression of CD148 and mapping them onto the AlphaFold CD18 structure reveals that they are not located within an FN3 domain but are surface-exposed and are positioned at either end of the flexible loop which extends from FN3-2 (Figure 5). Q276 lies within a short α-helix and replacement with proline would be associated with introduction of a kink in the helix.

A study exploring genes associated with hearing loss in Dutch families identified mutations in the PTPRQ gene associated with the condition [52]. One mutation was a nonsense mutation resulting in a truncated protein and the other was a missense SNP (R281G). PTPRQ plays a role in sensory hair cells where it is thought to be involved in cross-linking the shafts of the specialized cellular protrusions from these cells [53]. At a molecular level, the AlphaFold structure positions the R281 residue in the middle of a large flexible loop projecting from FN3-3 where it has the potential to make ionic, and other, interactions which would be lost on mutation to glycine. Alternatively, protein expression and localization may be affected.

## 6. Subgroup R4 (RPTPα, RPTPε)

The R4 subgroup of RPTPs includes RPTPα and RPTPε which are characterized by a short intrinsically disordered extracellular domain, which is heavily modified by N- and O- linked glycosylation, followed by a transmembrane domain and tandem intracellular phosphatase domains. No ligands have been reported for either RPTPα or RPTPε and its extracellular region may not be involved in ligand binding. A recent study of RPTPα provided a detailed mapping of site-specific glycosylation together with biophysical analyses [54]. The authors found that glycosylation shapes the disordered extracellular region creating what is described as a “floppy tentacle” rather than the familiar “bottle brush” structure of a typical mucin domain. They conclude that the heavy modification with negatively charged glycans favours the existence of RPTPα as a monomer at the cell surface.

At a functional level, RPTPα dephosphorylates the inhibitory Tyr-527 of the non-receptor tyrosine kinase Src leading to a conformational change, activation of the kinase and modulation of various growth factor and adhesion signalling pathways. Recently a novel mechanism regulating RPTPα activity has been proposed in which the glucose concentration that a cell is exposed to regulates RPTPα glycosylation and consequently its surface expression, phosphatase activity and Src activation, with implications for tumour growth [55]. Other mechanisms of regulation have been proposed, including a recently described mechanism involving RPTP clustering to enhance Src activation which promotes fibroblast-dependent arthritis and fibrosis [56]. Understanding the molecular basis for this RPTPα clustering would be of interest considering the Chien et al., study [54] indicating that an RPTPα monomer is the favoured conformation. AlphaFold structural predictions for the extracellular domain of this subgroup are unstructured, as expected.

## 7. Subgroup R5 (RPTPγ, RPTPζ)

The two R5 RPTP subgroup members—RPTPγ, RPTPζ—have an extracellular domain composed of a carbonic anhydrase-like domain (CAH), a single FN3-like domain, and a long, disordered spacer region followed by the transmembrane domain and tandem intracellular phosphatase domains. RPTPγ is widely expressed in tissues and cells including neuronal cells, haematopoietic cells, and hepatocytes, whereas RPTPζ exists mainly in the central nervous system on glial cells. The spacer region is heavily glycosylated in both RPTPs, and in RPTPζ alone it is also heavily modified by chondroitin sulfate. Several inhibitory ligands (pleiotrophin (PTPN), midkine and interleukin-34) have been identified for RPTPζ. The neural cell adhesion molecule contactin-1 (CNTN1) also binds to the CAH-like domain of RPTPζ and the four contactin-1 homologs CNTN3-6 bind to RPTPγ [57,58]. It has been proposed that the contactin molecules, which are glycosylphosphatidylinositol (GPI) anchored, adopt a bent conformation which places them parallel to the cell surface where they are able to interact either in *cis* or *trans* configurations with RPTPγ. The regulatory mechanism described for these RPTPs places them in a diffuse distribution at the cell surface in a monomeric active state due to the negatively charged chondroitin sulfate, or other glycosylation in the case of RPTPγ. Ligands with positively charged regions then bind to the negatively charged highly sulfated chondroitin sulfate chains resulting in reduced electrostatic repulsion, or a conformational change, which leads to formation of a dimeric, or oligomeric, and inactive form of the RPTP in the cell membrane. Based on the crystal structure of the tandem phosphatase domains of RPTPγ it has been proposed that the mechanism for this inhibitory signalling involves the formation of a “head-to-toe” dimer in which the D2 phosphatase domain from one molecule blocks the active site of the D1 domain in the pair [59]. This model has subsequently been confirmed in vitro and in vivo for RPTPζ [60]. Given that RPTPγ lacks chondroitin sulfate modifications, it will be interesting in the future to establish if, or how, RPTPγ ligands signal via this mechanism.

AlphaFold structural predictions (Figure 6) show the structured domains (CAH, FN3) and the spacer region which is significantly longer in RPTPζ than RPTPγ. 

## 8. Subgroup R7 (STEP, PTPRR)

The R7 subgroup includes striatal-enriched phosphatase (STEP) and PTPRR (also known as PCPTP1). Multiple transmembrane and cytosolic forms of these PTPs have been described in different species; however, there are no receptor-like forms of the closely related HePTP (*PTPN7*), and it is included with the non-transmembrane PTPs. The receptor-like form of PTPRR has a short extracellular domain which is predicted to be glycosylated followed by a transmembrane domain a kinase interaction motif (KIM) and a single intracellular phosphatase domain [61]. Interestingly, for the full-length form of STEP, referred to as STEP61 (based on its apparent molecular weight), two transmembrane domains and two proline-rich domains that would potentially enable binding to proteins with SH3 domains have been described [62]. This region is followed by a KIM sequence, a kinase-specificity sequence (KIS) and a single phosphatase domain. Both R7 subgroup PTPs, and HePTP, are negative regulators of the mitogen-activated protein kinases (ERK and p38) and binding specificity is determined by the KIM and KIS sequences. STEP61 has many other substrates including subunits of the ionotropic glutamate receptors (NMDA and AMPA), Fyn and Pyk kinases [62]. The short AlphaFold extracellular domain structures are not presented here.

## 9. Subgroup R8 (IA-2, IA2β)

The R8 subgroup of receptor PTPs, IA-2 (ICA512, *PTPRN*) and IA2β (phogrin, *PTPRN2*) have a large extracellular domain a transmembrane domain and a single intracellular pseudo-phosphatase domain, which in the case of IA2β has weak activity towards inositol phospholipids [63]. They are mainly expressed in neuroendocrine cells, including insulin-secreting pancreatic β-cells, where they are involved in granule cargo storage, exocytosis, and cell proliferation. They are prominent type 1 diabetes autoantigens, and autoantibody detection is used as a predictive marker of type I diabetes. Proteolytic processing of the precursor protein by furin-like convertases cleaves off a large N-terminal fragment leaving a transmembrane protein of approximately 60 kDa. The structure of the mature extracellular domain has been solved and is related to the SEA (sea urchin sperm protein, enterokinase, agrin) domain of mucins [64]. This region is initially orientated towards the lumen of the granule and then towards the extracellular space following granule secretion. AlphaFold structural predictions for the mature extracellular region are shown in Figure 6.

## 10. Conclusions

Structural analysis of extracellular domains of the RPTP family reveals the diverse conformations that these molecules adopt to fulfil their signalling roles. AlphaFold-computed structures suggest that all R2B subgroup RPTPs adopt an extended conformation while all R2A RPTPs adopt a highly flexible conformation which correlates with available experimental structural information for specific family members. Where experimental structure information is limited, such as with the R3 RPTP family, AlphaFold models provide fascinating new insights into the potential conformations of these molecules and position of disease-associated missense mutations. Extracellular regions from this subgroup are not uniform in the conformations they adopt. Both CD148 and SAP-1 are predicted to adopt an extended conformation while PTPRO has a bent L-shape and other subgroup members (VE-PTP and PTPRQ) are twisted, reflecting their high flexibility. The extended conformation of CD148 is in accordance with its documented role in size exclusion at regions of cell–cell contact, and by analogy, SAP-1 may perform a similar role, albeit at different cellular locations. In contrast with previous models for R3 RPTPs which suggest that the extracellular region of these molecules is formed solely of FN3 domains, AlphaFold predicts that CD148, PTPRO and PTPRQ have one or more unstructured loops in place of some of these domains. The functional significance of the predicted loops is unknown, but it is interesting to note that in the case of CD148 and PTPRQ, disease-associated missense SNPs are found at either end (CD148) and near the middle (PTPRQ) of one loop, rather than within an FN3 domain as previous models suggested, providing a possible alternative insight into disease mechanisms. AlphaFold can be used to visualise the position of a mutation within a 3D protein structure, but it should be noted that it is not able to predict the effect of missense mutations on the 3D structure of a protein and is not expected to produce an unfolded protein structure given a sequence containing a destabilising point mutation [65].

The highly accurate protein structure predictions generated by AlphaFold do not replace experimental structural studies [66], as they do not take into account post-translational modifications such as glycosylation, or protein–protein interactions, ligands and the plasma membrane; nonetheless, they provide a valuable foundation for future experimental studies investigating the function of these proteins.

## Figures and Tables

**Figure 1 ijms-25-00820-f001:**
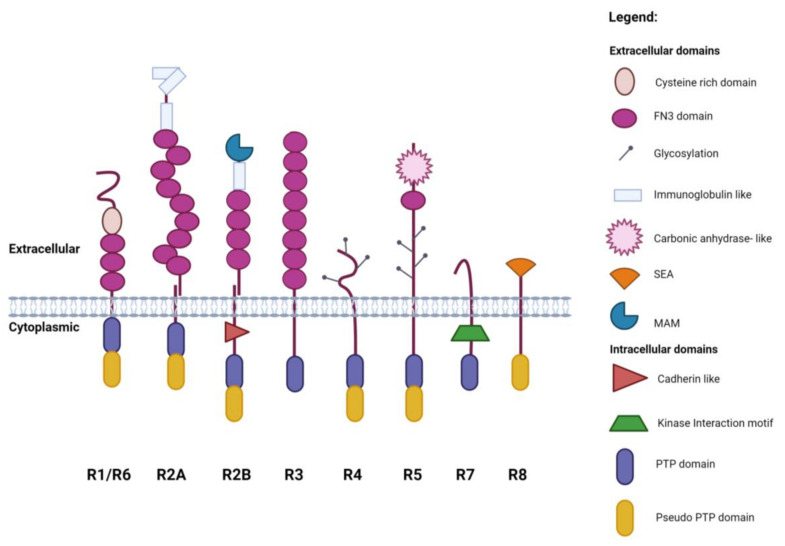
Schematic representation of the RPTP family. Created with BioRender.com.

**Figure 2 ijms-25-00820-f002:**
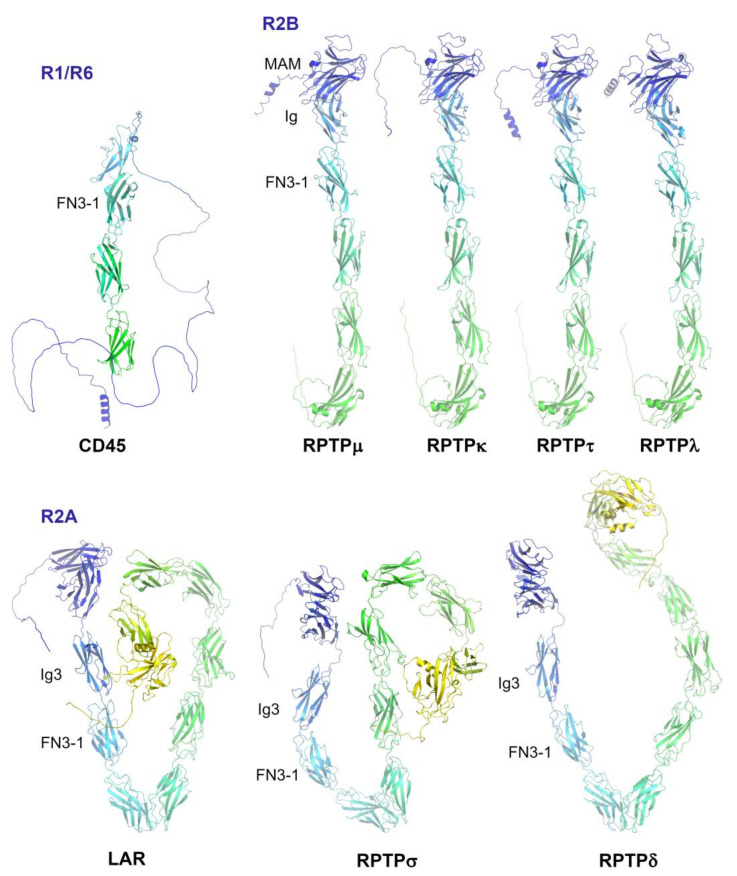
AlphaFold structural predictions of the extracellular domain for R1, R2A and R2B subgroup RPTPs. Images were generated using PyMol with structures coloured by spectrum (N-terminus, blue; C-terminus, yellow). Human structures were downloaded from: https://alphafold.ebi.ac.uk/ accessed on 10 November 2023.

**Figure 3 ijms-25-00820-f003:**
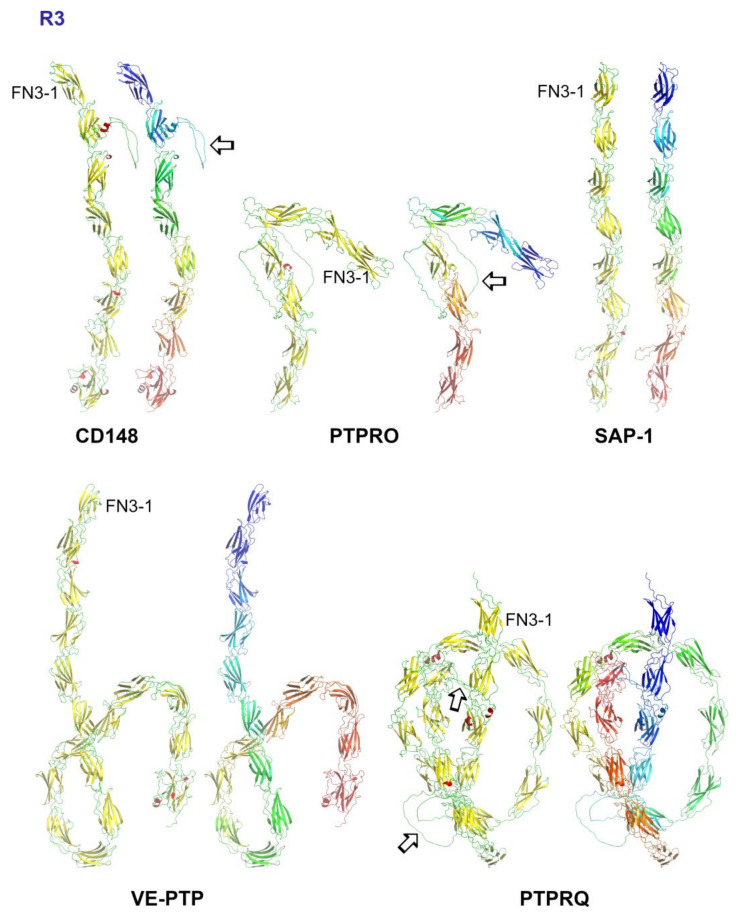
AlphaFold predicted structures for the extracellular domain of R3 subgroup PTPs. Images were generated using PyMol with colouring by secondary structure (helix, red; sheet, yellow; loop, green) and by spectrum (N-terminus, blue; C-terminus, yellow/red). For clarity, the sequence preceding FN3-1 and from the predicted transmembrane domain onwards is not shown. Unstructured loops are indicated by an arrow.

**Figure 4 ijms-25-00820-f004:**
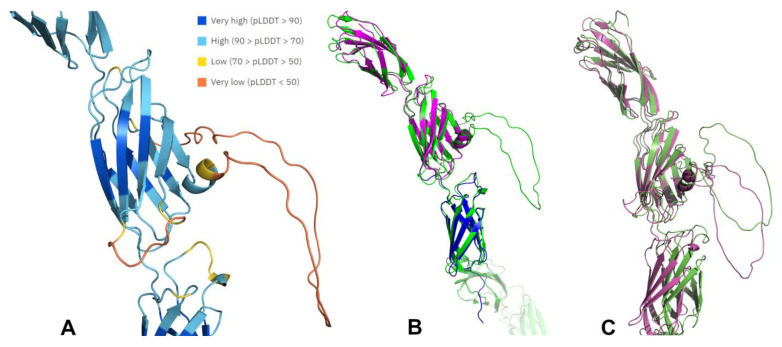
The unstructured loop of CD148 extracellular domain. Images were generated using PyMol (**A**) Part of the AlphaFold predicted extracellular domain CD148 structure coloured by pLDDT colours (credit Konstantin Korotkov). (**B**) Superimposition of the AlphaFold predicted CD148 structure FN3-1 to FN3-3 (AF-Q12913-F1, green) with an X-ray crystal structure of CD148 FN3 domains 1 and 2 (PDB: 7U08, magenta) and the NMR solution structure of FN3-4 of CD148 (PDB: 2DLE, blue). (**C**) Superimposition of a section of the CD148 AlphaFold structure (green) and a Colabfold structure generated without templates (magenta). ColabFold (v1.5.2) was installed from https://github.com/YoshitakaMo/localcolabfold on a Linux system and the structure was generated using amino acids 36–975 of human PTPRJ (Q12913) with the command “colabfold_batch colabfold_input outputdir/” accessed on 7 November 2023.

**Figure 5 ijms-25-00820-f005:**
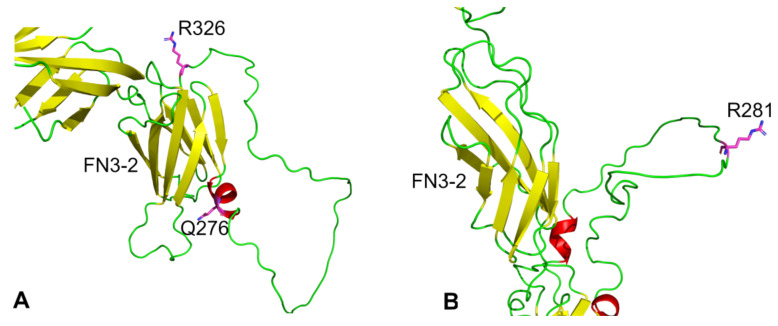
Location of missense SNPs in AlphaFold structures of CD148 and PTPRQ. Images were generated using PyMol with colouring by secondary structure (helix, red; sheet, yellow; loop, green) (**A**) Structure of a section of the AlphaFold CD148 extracellular domain showing the position of the SNPs (Q276P and R326Q). Ribbons for other domains have been removed for clarity. (**B**) Structure of a section of the AlphaFold PTPRQ extracellular domain showing the position of the SNP (R281G).

**Figure 6 ijms-25-00820-f006:**
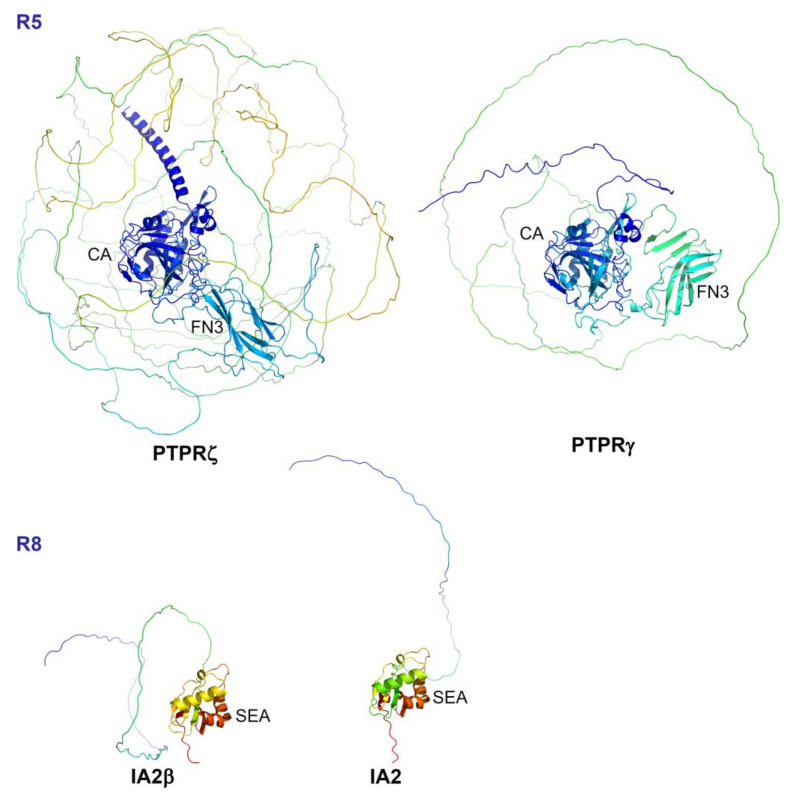
AlphaFold structures of the extracellular region of R5 and R8 subgroup PTPs. Structures are coloured by spectrum (N-terminus, blue; C-terminus, yellow/red). The structured carbonic anhydrase-like (CA) domain, single FN3-like (FN3) domain and long disordered spacer region are shown for the R5 group. SEA (sea urchin sperm protein, enterokinase, agrin) module.

**Table 1 ijms-25-00820-t001:** Predicted FN3 domains and loops in R3-PTPs based on UniProt and AlphaFold.

**Name**	Gene Name	UniProt	UniProt (Prosite) FN3 Domains	AlphaFold FN3 Domains	Loops, Length and Position
CD148	*PTPRJ*	Q12913	9	8	N278–D327 (49aa) at FN3-2
VE-PTP	*PTPRB*	P23467	17	17	none
GLEPP-1	*PTPRO*	Q16827	8	7	N234–P342 (108aa) at FN3-3
SAP-1	*PTPRH*	Q9HD43	8	8	none
PTPRQ	*PTPRQ*	Q9UMZ3	18	18	L254–N298 (44aa) at FN3-3 T491–E569 (78aa) at FN3-5

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
