# Peer review of "Analysis of Receptor-Type Protein Tyrosine Phosphatase Extracellular Regions with Insights from AlphaFold"

_ijms, 2024, doi:10.3390/ijms25020820_

Round 1

Reviewer 1 Report

Comments and Suggestions for Authors

In this review, structural and functional aspects of the the extracellular region of receptor-type protein tyrosine phosphatase (RPTP) are discussed and aimed to be corroborated by AlphaFold predictions. While the existing structural knowledge on 8 subgroups is well reviewed and presented, the same cannot be said for the findings from AlphaFold, as promised in the title. This part is not presented in the form of a review, but rather appears to be a novel study performed by the authors. Nevertheless, this part lacks a critical
evaluation of the results. No details on the computational part are given. Also missing is a general discussion of the predictions including "the fascinating new findings", especially in the concluding section.

Minor points:

Once the term AlphaFold2 is abbreviated to AF2, it should be used strictly throughout the entire manuscript.

Repeated "to" in the last line of page 9.

Author Response

We thank the reviewer for their comments.

Findings from AlphaFold lacks critical evaluation – We have extended the conclusion to acknowledge the limitations of AlphaFold-generated structures and make the point, as published by others, that the AlphaFold computed structures should be viewed as exceptionally useful hypotheses that require experimental structural studies to verify details.

No details of computational part are given –The only computed structure used in the article was generated using Colabold (Fig 4C) and we have now included details of how this was executed in the Figure legend. Other AlphaFold structures were downloaded from EBI (https://alphafold.ebi.ac.uk/).

General discussion on predictions and critical evaluation of results:  Lines 256 – 305 provide a critical evaluation of the findings from the AlphaFold2 structures for R3 subgroup PTPs.  We have expanded the conclusion to clarify the fascinating new findings that we refer to in the text.

Minor points

We have removed the abbreviation AF2 and now refer to AlphaFold throughout the manuscript, and title, rather than AlphaFold2 to avoid confusion and to be consistent with other published work.

Typographical errors have been corrected

Reviewer 2 Report

Comments and Suggestions for Authors

In this article, the authors provide a comprehensive overview of receptor-type protein tyrosine phosphatases (RPTPs), highlighting their involvement in a diverse range of physiological functions. This comprehensiveness helps readers understand the broad significance of RPTP in cellular processes. The review effectively highlights the diversity of the extracellular regions of RPTP. Metaphorical descriptions such as “stiff rods” and “flexible tentacles” make the complex nature of RPTP accessible to a wider audience, helping to convey scientific concepts. The Review combines existing experimental structural knowledge with insights derived from AlphaFold's structural predictions. This integration strengthens the analysis, potentially providing a more complete understanding of the structure and function of RPTP.

The Review can be published in International Journal of Molecular Sciences. There are however several recommendations:

1. Abstract: Please, provide the description of the Review chapters and underline main analytical findings and novelty. It also would be good to make conclusion explaining the importance and medical relevance of the Review.

2. Introduction: Please explain why it is so important to investigate the structural features of RPTPs and how it furthers understanding of biological processes involved in diseases.

3. It would be good to prepare more detailed description of functional significance of various groups of RPTPs and to prepare a color figure unifying the description for all groups.

4. Please, extent the conclusion by outlining the main findings, novelty abd the importance of the knowledge presented in the Review for biology and medicine.  

Author Response

We thank the reviewer for their comments.

1 We have now highlighted some of the main findings and novel aspects in both abstract and conclusion and stated that many RPTPs are dysregulated in disease and are drug targets.  Throughout the review in the section for each subgroup we have provided detail relevant to importance, physiological and medical relevance.

2 We have added further sentences in the introduction to explain why it is important to understand RPTPs structural features for investigations of disease and drug development.

3 Descriptions of functional significance are included within the sections for each subgroup or we have referred to reviews where this has been covered recently.  I do not know what sort of colour figure unifying the description for all groups is being requested.  Figure 1 provides a colour schematic showing all of the extracellular domains.  This has been updated relative to previous review articles to incorporate structural insights discussed in the review.

  1. We have extended the conclusion as suggested.

Round 2

Reviewer 1 Report

Comments and Suggestions for Authors

I have no further criticism.

Author Response

Thanks for the review, we appreciate it

Reviewer 2 Report

Comments and Suggestions for Authors

The Authors addressed all my comments.

The Article can be published.

Author Response

Thanks for the review, we appreciate it